# The evolution of convex trade-offs enables the transition towards multicellularity

Joana P. Bernardes[1,5], Uwe John [2,3], Noemi Woltermann[1], Martha Valiadi [1,6], Ruben J. Hermann [4] & Lutz Becks [1,4✉]

The evolutionary transition towards multicellular life often involves growth in groups of undifferentiated cells followed by differentiation into soma and germ-like cells. Theory predicts that germ soma differentiation is facilitated by a convex trade-off between survival and reproduction. However, this has never been tested and these transitions remain poorly understood at the ecological and genetic level. Here, we study the evolution of cell groups in ten isogenic lines of the unicellular green algae *Chlamydomonas reinhardtii* with prolonged exposure to a rotifer predator. We confirm that growth in cell groups is heritable and characterized by a convex trade-off curve between reproduction and survival. Identical mutations evolve in all cell group isolates; these are linked to survival and reducing associated cell costs. Overall, we show that just 500 generations of predator selection were sufficient to lead to a convex trade-off and incorporate evolved changes into the prey genome.

[1] Community Dynamics Group, Department of Evolutionary Ecology, Max Planck Institute for Evolutionary Biology, D-Plön, Germany. [2] Alfred Wegener Institute Helmholtz Centre for Polar and Marine Research, Bremerhaven, Germany. [3] Helmholtz Institute for Functional Marine Biodiversity, Oldenburg, Oldenburg, Germany. [4] Limnological Institute University Konstanz, Aquatic Ecology and Evolution, Konstanz, Germany. [5] Present address: Institute of Clinical Molecular Biology, Kiel University, Kiel, Germany. [6] Present address: Institute for Molecular Biology and Biotechnology, Foundation for Research and Technology Hellas, Heraklion, Crete, Greece. ✉email: lutz.becks@uni-konstanz.de

The transition from unicellular to multicellular life occurred multiple times independently among living species (~25–30 times)[1]. The initial evolutionary step involves the growth of undifferentiated cells in groups (hereafter: cell groups; for a review see, e.g., ref. [2]). Previous work has described at least two mechanisms, as well as a combination, leading to the evolution of multicellularity: (1) aggregation of genetically diverged free-living cells like in cellular slime molds and Myxobacteria[3–5] or (2) cellular adhesion of clones unable to fragment after cell division like in the *Chlamydomonas* green-algae[6–8] or in *Saccharomyces* yeasts[9]. One condition under which the evolution of cell groups has been observed repeatedly, is in response to predation by filter feeders (e.g., unicellular eukaryotes and zooplankton[6,10–13]). Growth in cell groups increases handling time and/or decreases ingestion rates of the predator and thus survival of the cell groups, as larger cell groups are less frequently preyed upon. Increased survival in the presence of predators is, however, often associated with lower reproduction, due to lower resource uptake[14], loss of motility when daughter cells do not separate after cell division[15,16] and reduction of photosynthetic rate with increased sinking out of the euphotic zone[17–19]. In order to understand the conditions and constraints for the evolution of cell groups and subsequent steps towards multicellularity, with separation into soma cells (cell groups) and germ cells (single cells starting the next generation), we need to recognize the evolution of traits and their trade-offs in response to predation. This is because the trade-off between reproduction and defense against consumption by predators can determine the likelihood of the evolution towards multicellularity[20].

The evolution of clonally derived cell groups in response to predation has been reported previously[6,21,22], but the curvature of the trade-off (i.e., convex, linear or concave) has not yet been tested. The trade-off curvature is, however, important, because the shape of the trade-off curve can constrain adaptive responses to selection. Several studies highlighted the importance of the curvature of the trade-off curve[23], for example, life-history trait evolution in general[24], the evolution of defense in the presence/absence of coevolving predators[25] and in the coexistence of species or genotypes[26]. For the evolution of multicellular life, no cell groups are expected to evolve when an increase in reproduction leads initially to only small changes in survival (Fig. 1a, dashed line = concave trade-off curve)[27]. The evolution of cell groups is, however, more likely when an increase in survival comes at a relative high cost of reproduction, where survival evolves first followed by an increase in reproduction (Fig. 1a, solid line; convex trade-off curve). The second step in the evolutionary transition toward multicellularity, the soma-germ separation, is more likely expected when the trade-off shifts towards convex curvature with increasing group size and when the fitness trade-off creates a covariance effect[20]. The covariance effect at the group level allows a higher group fitness compared to the average fitness of the component cells (soma and germ) and could thus lead to a situation where the benefits eventually outweigh the reproductive costs of cell groups.

Traits are often influenced by factors such as phylogenetic origin or cell size, which limit comparisons of trait distributions between related species and trait correlations[28]. Thus, trade-off curves are most informative when comparing estimates from intraspecific clones although clonality is in general not a requirement for the evolution of multicellularity[29]. Here, we use experimental evolution to evolve intraspecific trait diversity in ten cell lines representing distinct genotypes of the unicellular green-algae *Chlamydomonas reinhardti*. This allows us to examine phenotypic variation and assess the impact of genomic variants in parallel cultures, without confining selection to one genotype. By exposing each line to the predator *Brachionus calyciflorus*, we test for the likelihood of evolving growth in cell groups under predation. We then evaluate the costs associated with growing in cell groups and the form of a trade-off. Finally, we set out to identify the underlying genetic changes. Growth in groups of *C. reinhardtii* has been observed under a number of different conditions independent of the presence of consumers[21,30,31]. To account for this, we grow each strain of *C. reinhardtii* in both the presence and absence of the predator with all other experimental conditions remaining identical. After ~500 generations growing in the presence/absence of the predator, we isolate 8–12 algal clones from each selection line and classify the isolates into different morphotypes (motile single cells, non-motile single cells, motile and non-motile single cells, cell groups, cell groups and motile single cells, cell groups and non-motile single cells) and measure their survival and reproduction. All isolates are tested under standardized conditions and at least 20 generations after isolation from the selection experiment to ensure the phenotypes observed here are heritable. Further evidence for heritability comes from comparing the gene expression patterns of single cells and cell groups under different environmental conditions. Finally, we use whole-genome sequence information for nine pairs of evolved single cells and cell groups to identify shared mutations for the different morphotypes (single cells, cell groups) and develop a graphical model summarizing the genomic changes involved in increasing survival and its associated costs in cell groups of *C. reinhardtii*.

## Results

**Cell groups and trade-off.** Exposure to predators led to an increase of algal cell groups with high survival, despite consequences for their reproduction. We found significantly more cell groups, where cells adhered to each other after cell division (Fig. S1), among the isolates from the predation lines compared to isolates from non-predation lines (predation 49 out of 97 algal clones tested, no predation 15 out of 112 tested morphotypes (Chi-square: $\chi^2 = 26.07$, $p = 3.29*10^{-7}$; Fig. 1b)). With predation, all ten isogenic lines evolved groups, without predation only four out of ten lines. Growth in groups increased survival (i.e., rotifer growth rate was significantly reduced for cell groups; linear mixed effect model (LME) with strain as random effect: $F_{1,202} = 6.40$, $p = 0.012$; Fig. 1c and Fig. S2) but reduced average algal growth rate compared to single cells (LME: $F_{1,202} = 15.82$, $p = 9.74*10^{-5}$; Fig. 1c).

After confirming the evolution of cell groups, we established the trade-off curvature for cell groups and single cells. We fitted a power law model of the form $y = a*x^b$ to the data, where $y$ is the algal growth rate, $x$ is the rotifer growth rate, $b$ the parameter determining the shape of the curve (for $b < 0$ convex, for $b > 0$ concave) and $a$ is the minimum algal growth rate. For cell groups and single cells in both treatments, the shape parameter $b$ is negative and significantly different from zero (single cells: $b = -0.34 \pm 0.1$; df = 3, $F = 97.7$, $p < 2.9*10^{-16}$; cell groups: $b = -0.56 \pm 0.24$; df = 3, $F = 37.7$, $p = 6.38*10^{-8}$; significant levels determined by model comparisons with and without exponent; Fig. 1c). This confirms the predicted convex trade-off between survival and reproduction. Nevertheless, the curves for single cells and cell groups differ, with cell groups having a stronger trade-off for reproduction and survival. For example, for a given level of survival (horizontal line in Fig. 1c), single cells have higher reproduction and for a given level of reproduction (vertical line in Fig. 1c), cell groups have a higher survival. When comparing the trade-off between cell groups and single cells within the two treatments, we found a significant trade-off only for cell groups for isolates from the predation lines (cell groups: $b = -0.52 \pm 0.28$; df = 3, $F = 21.33$, $p = 3*10^{-5}$; single cells: df = 3,

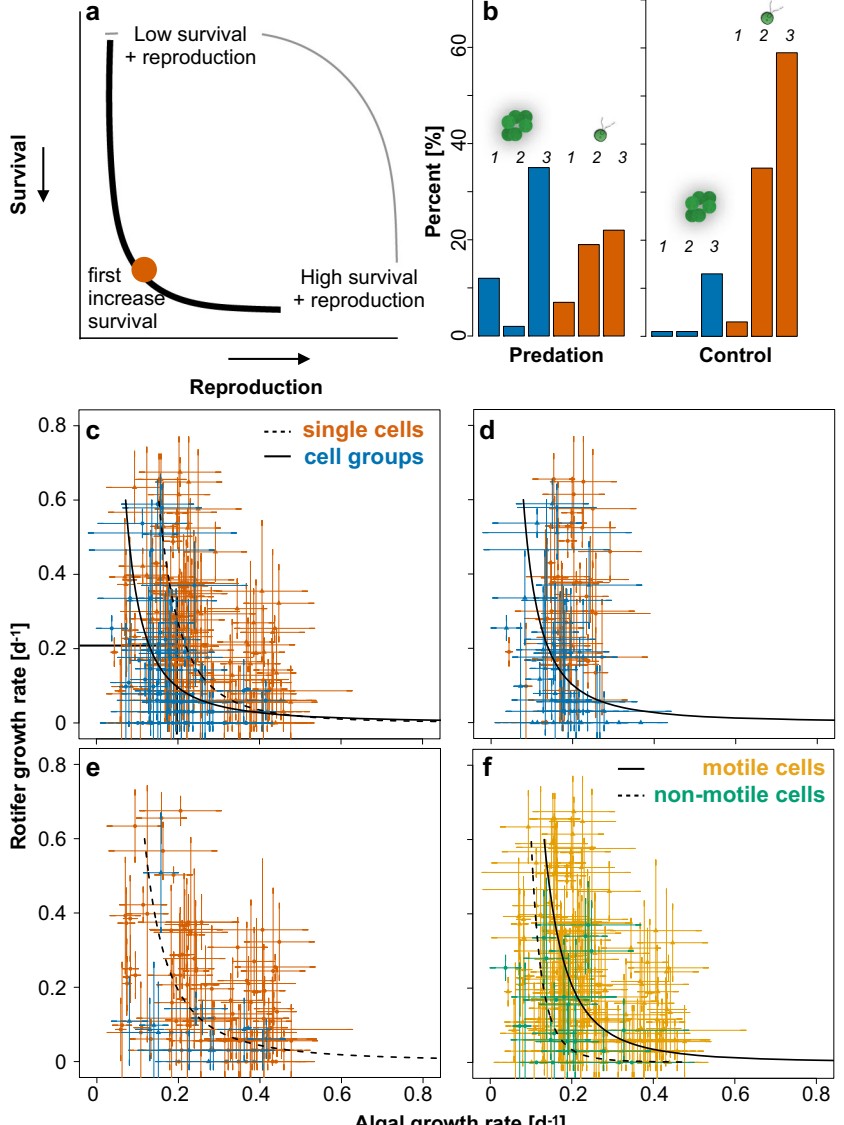

**Fig. 1 Selection by predatory rotifers lead to the evolution of cell groups with higher survival but lower growth rate of algae. a** Conceptual framework for the evolution of clonal cell groups as a function of the trade-off between reproduction and survival. The convex curvature of the trade-off (increase in survival first) with the orange dot is the one predicted to select first for cell groups followed by the evolution of soma-germ specialization. The concave curve (increase in reproduction first) on the other hand, is predicted to select for fast growing single cells but no further steps towards multicellular life. Modified from ref. [20]. **b** Frequency of clone morphologies of isolates classified into single cells (red) with the subcategories no-motile single cells (1), some motile single cells present (2), majority motile single cells (3), as well as into cell groups (blue) with the subcategories: groups with no single cells (1), groups with non-motile single cells (2) and groups with motile single cells (3). There were significantly more cell groups selected by predation. **c** Growth rates of algal isolates and rotifer growth rates when feeding on algal isolates (higher defense of algal isolated for lower rotifer growth rate = higher algal survival). Blue points show data for isolates in cell groups, red triangles for single cells. Black solid line represent predicted trade-off curve for cell groups, black dashed line for single cells (see main text). **d** Subset of data from **c** with isolated from the predation-treatment. **e** Subset of data from **c** with isolated from the predation-treatment. **f** The same data as in **c** but colors indicate motile (yellow) and non-motile single cells/cell groups (green). For **c**–**f** shown are means of three replicated measurements per isolate (arrow bars: standard deviation of replicated measurements per isolate) and only significant trade-off curves. For more details see main text. Source data are provided as a Source Data file.

$F = 0.0128$, $p = 0.9105$; Fig. 1d) and only for single cells in the no-predation lines (cell groups: too few points to fit model; single cells: $b = -0.47 \pm 0.12$; df $= 3$, $F = 108.07$, $p < 2.2*10^{-16}$; Fig. 1e). Potential costs for *Chlamydomonas* growing in cell groups include the potential for self-shading, deflagellation, loss of motility and/or sedimentation. Indeed, we found a significant lower frequency of swimming cells across all isolates growing in groups ($\chi^2 = 6.85$, df $= 1$, $p = 0.009$; Fig. 1b). Trade-off curves were also more convex for motile and non-motile isolates (motile: $b = -0.39 \pm 0.12$; df $= 3$, $F = 107.1$, $p < 2.9*10^{-16}$; non-

motile: $b = -0.24 \pm 0.32$; df $= 3$, $F = 7.45$, $p = 0.013$; significant levels determined by model comparisons with and without exponent, Fig. 1f), but non-motile isolates had a lower growth rate for similar survival rates.

**Gene expression patterns**. We further tested the heritability of the evolved phenotypes by comparing the gene expression patterns for one-pair single cells from the no predation lines (majority motile single cells) and evolved cell group from the predation line (groups with no single cells) using RNA-seq of the

wildtype strain cc1009 (Table S1). We grew isolates under different environmental conditions in replicated chemostats (continuous culture systems, $n = 3$ per condition): with predation by rotifers (+R), with reduced nitrogen concentration to resemble low-resource conditions (−N; we halved the concentration of N to 40 µM) or without any of the two stressors that the populations experienced in the selection experiment (control C). After 5 days in these conditions, we harvested the algae from the chemostats and determined population sizes and cell groups (Fig. S3). We correlated the normalized transcriptome counts using the Spearman rank method. Overall, cell groups displayed a more characteristic, i.e., less variable, transcription profile independent of the environmental conditions as we found higher correlation scores between treatments (average rho = 0.97) growing in cell groups than single cells (average rho = 0.93, Fig. 2a). The differences between cell groups and single cells were additionally supported by the distance separating the two groups in a non-metric multi-dimensional scaling (MDS), with cell groups showing higher similarities than single cells (Fig. 2b).

Transcriptome differences between cell groups and single cells revealed 76 differentially expressed genes (57 upregulated and 19 downregulated in cell group algae, batch corrected for the treatments) across treatments (Fig. 2c). We focused on the genes that were differentially expressed between cell groups and single cells, separated them into genes that were down- or upregulated in cell groups and identified an enrichment for the following pathways (pathway enrichment based on protein identity and FDR-based analysis; Fig. 2d): genes that were downregulated in cell groups were enriched for RAF/MAP kinase cascade, FLT3 signaling, signaling by receptor tyrosine kinases and EPH-Ephrin signaling, and upregulated genes for voltage gated potassium channels, TRP channel, G-protein mediated events, signaling by receptor tyrosine kinases and PLC beta mediated events.

We further compared the differentially expressed genes to a recent study with *C. reinhardtii* and rotifers[32] looking at the gene expression over time (3, 6, 9, 12, and 48 h) of an evolved isolate that is characterized by unicellular propagules from which multicellular clusters grow. Specifically, we compared differentially expressed genes from ref. [32] detected for each life cycle phase (growth lag-phase → unicellular propagule → multicellular cluster) to our dataset of differential expressed genes identified in cell groups when compared with single cells. We found fewer genes in common between the lag-phase of the life cycle (3 h) and identified in our cell group selection (352 genes), while several more genes were in common at the life cycle phase of development of multicellular clusters (686 genes at 12 h and 731 genes at 48 h). In total there were 174 genes that were differentially expressed both in cell groups in our dataset and in all multicellular induction phases in ref. [32]; 64 of those genes were downregulated and 110 upregulated in cell groups and throughout the multicellular phases while 110 were upregulated in cell groups and throughout the multicellular phases (see gene list Supplementary Data 1).

**Functional genomic changes**. To identify genomic changes that are potentially involved in growth in cell groups and the trade-offs, we whole-genome sequenced one-pair of isolated algal clones growing in groups from the predation lines and as single cells from the non-predation lines from nine isogenic strains (total nine isolates in cell groups, nine single-celled; see Table S2 for morphology; isolates from cc2531 were not included due to insufficient DNA quality for sequencing). We identified positions along the genome that contained variants (single-nucleotide polymorphism: SNPs and small indels) compared to the reference

genome[33]. As we compared all isolates to the same reference genome, we cannot distinguish whether a variant evolved de-novo in the cell groups or in the single cells. We found a total number of 156,872 ± 84,970 variants for cell groups and 154,314 ± 47,240 variants for single cells. We categorized these as: common in all clones, unique to cell groups and unique to single cells. A higher number of these was shared across cell groups (181; Supplementary Data 2) than across single cells (83). 59 variants unique to the cell groups were in coding regions (single cells: 23) with 24 being predicted to lead to an amino-acid change (single cells: 12). Variants in the 3' and 5' regions with amino-acid change, possibly involved in gene regulation, were rather similar shared in the cell groups (3'prime: 13, 5'prime: 15) than in single cells (3'prime: 21, 5'prime: 7).

Five of the shared variants with amino-acid changes from the isolated cell groups were within the gene *DUF3707* (Cre01.g049826, Cre06.g290676). This gene belongs to the family of pherophorins, which are extracellular matrix glycoproteins. Other mutations were found in genes encoding for other glycoproteins (Cre10.g431050), proteins involved in excretion (Cre06.g290676), and a 5' region regulating genes for extracellular matrix proteins (Cre12.g522650). Mutations in genes coding for proteins associated with nutrient uptake (Cre12.g491600), cytosolic viscosity (Cre06.g266300) and 3' regions for genes involved in the stress response to $CO_2$ stress (Cre10.g452800) are possible responses to the costs of growing in cell groups. All other variants with predicted amino-acid changes were found in genes, 3' or 5' regions that deal with protein and transcription regulation, but these functions were not unique to variants in isolates growing in cell groups. We further analyzed the variants of interest by comparing their transcript expression between single cells and cell groups. There were no significant differences between the two groups (Fig. S4), however, we cannot discard the cumulative effect of small expression changes.

## Discussion

One of the central steps in the evolution towards multicellularity is the differentiation of somatic and germ cells whereby the cells of the body (soma cells; cell group) must play an ecological role (e.g., predation avoidance, motility that increases access to resources) while simultaneously initiating the next generation of bodies (germ-like cells; single cell). The differentiation into germ and soma cells can be favored due to an inherent fitness trade-off between survival and fecundity[34]. Theoretical work analyzing this trade-off showed that the survival-fecundity trade-off could lead to the separation of soma and germ cells when the costs of increasing cell group sizes become larger than the benefits of being in cell groups or aggregates[27,34]. Here, we tested experimentally the evolution of growth in cell groups by evolving populations of the unicellular *C. reinhardtii* in different environments, characterizing their phenotypes and finding mutations that are likely involved in the observed phenotypes. Specifically, we checked for the curvature of the trade-off between survival in the presence and absence of a predator and reproduction of the algal cells, as the curvature is predicted to determine the likelihood that cell groups can evolve further to specialize into soma and germ line cells[20]. We found that isolates of algal clones that were selected under rotifer predation were more likely to grow as cell groups and displayed a decrease in growth rate and increase in the defense level against consumption by the predator, i.e., survival. We found a convex trade-off curve between reproduction and survival as predicted. The majority of cell groups we observed were growing as groups and they had increased survival in the presence of a predator. We found a smaller percentage of isolates that grew as a mix of groups and single cells, but it is

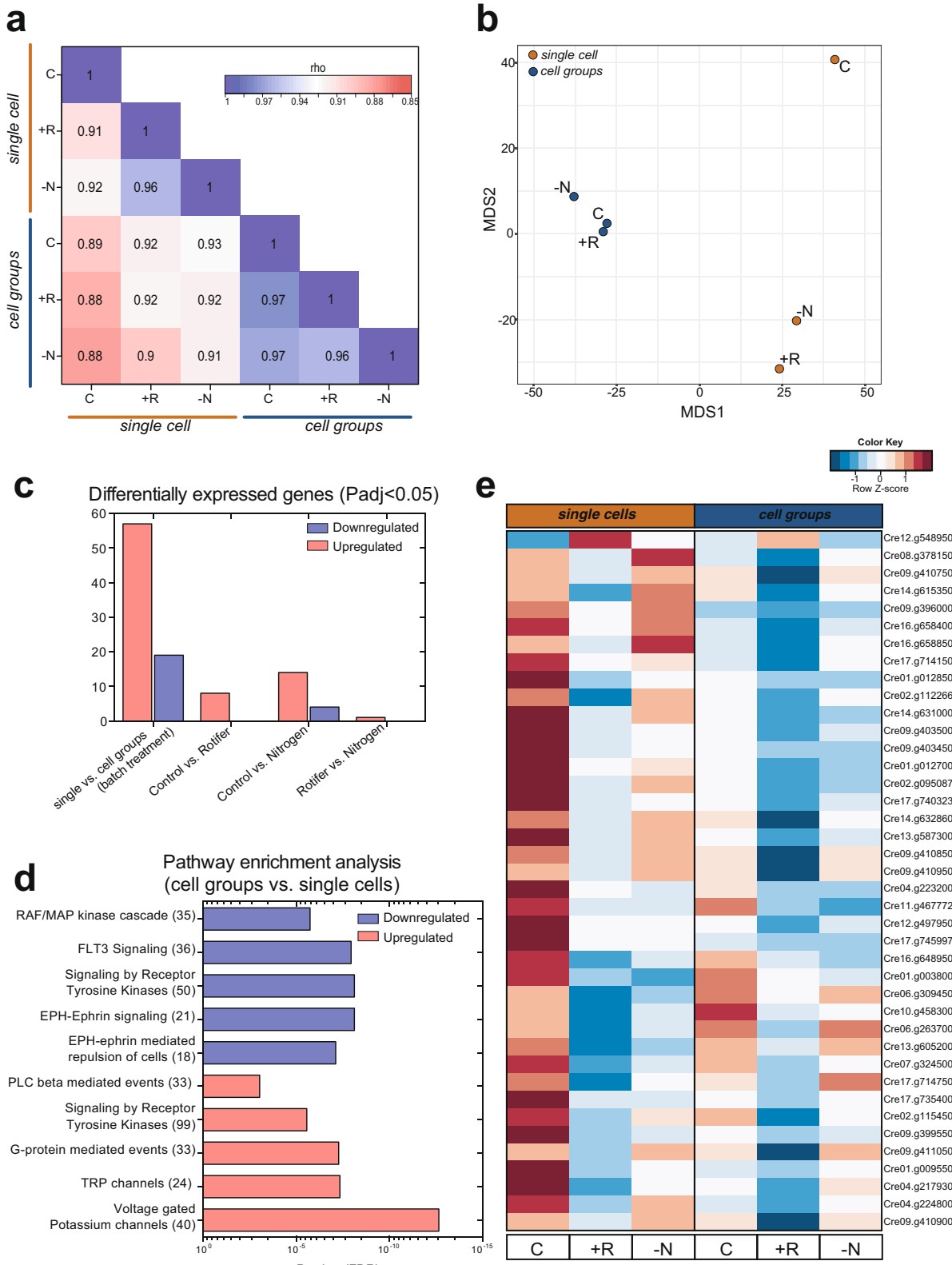

**Fig. 2 Heritable transcriptome of single cells and cell groups. a** Spearman correlation plot and **b** MDS analysis of algal transcription profiles for single cells (red) and cell groups (blue) clones separated by control (C), rotifer predation (+R) and, low-nitrogen (-N) treatments. **c** Plot depicting differentially expressed genes for multiple comparisons. **d** Significantly enriched pathways in based on the differentially expressed genes of discovered for cell groups and **e** heatmap referring to the top40 differentially expressed genes between cell groups and single cell clones. Source data are provided as a Source Data file (**c**).

unclear whether these represent different life stages of a cell group or parts of cell groups as observed in other studies[8,32]. As we focus in our study on the curvature of the trade-off curve, our experimental environments were not designed for testing for ecological scaffolding and separating fitness benefits of the soma (cell group) and germ cell-like cells (single cells)[35–37]. Whether these mixed isolates would provide a fitness benefit in environments where selection for germ and soma are temporally and/or spatially separated or whether the fitness of the group is larger than the average fitness of the germ and soma cells[20], needs further investigations. The observed lower growth rate for a given survival rate of non-motile cells compared to motile cells and selection in static cultures in our experiments suggest that motility could be beneficial for accessing nutrients in the planktonic phase and/or access to light.

*C. reinhardtii* probably never had a multicellular ancestor and resembles the unicellular ancestor of the volvocales[38], making it the best candidate species for our experiments. The volvocine algae encompass a continuum of transitional types towards multicellularity ranging from a unicellular lifestyle (*C. reinhardtii*), to life in cell groups (*Gonium pectorale, Pandorina morum*), to a multicellular lifestyle with a developmental program, partial and full soma-germ differentiation and an increased extracellular matrix (*Pleodorina starrii, Volvox carteri*). In our experiment, *C. reinhardtii* evolved to grow in groups, where cells embedded in an extracellular matrix and daughter cells fail to separate fully after cell division. It is possible that cells were kept together by the extracellular matrix or failure of the cell septum to form. The observed genomic changes suggest the former (see below and Fig. 3) but further analyses are required to identify the underlying mechanism. The difference in gene expression between cell groups and single cells was prevalent, and independent of the environment. Indeed, we found common expression changes between cell groups and those previously described when transition to multicellularity life cycle was induced[31], hinting to potential commonalities in impacted metabolic pathways. The conserved transcriptome of single cell versus cell group and the experimental set-up we used to measure all the phenotypes (i.e., after >20 generations in a common environment) support that the phenotypic changes were heritable, although we currently cannot distinguish between epi-genetic and genetic changes. The evolution of heritable growth in groups in response to predation has been observed in previous studies with *C. reinhardtii*[6,13] and other green-algae[10]. Cell group formation in response to predation has been observed in some other studies as a phenotypically induced responses[19,21] rather than a heritable phenotype. The underlying process for cell group formation could account for the differing observations between ours and other studies. While cell groups in the present study are the result of growth in groups where the daughter cells fail to separate after cell division (Fig. S2), in other studies cells actively aggregated for the formation of the group of cells[19,39].

The genomic changes involved in the initial step of growth in cell groups are mostly unknown but could provide important insights into the evolutionary route to cell groups and the trade-off between survival and reproduction. In our study we only addressed genetic, but not epi-genetic, changes. However, we only focused on changes that were shared within, but not between, single cells or cell groups, to identify patterns that were independent of the specific initial genotype in the selection experiment. Previous work using comparative genomics suggests that changes in genes involved in cell cycle regulation and synchronization are associated with the evolution of multicellularity[40]. These observations were supported by an experimental evolution study showing significant changes in gene expression in genes related to cell cycle regulation[32]. Other experimental evolution

studies identified mutations in genes associated with the lack of separation of daughter cells[41,42] and thus the genomic changes associated with increases in cell group size. We found a large number of shared sequence variants (SNPs) across cell groups indicating that the changes are involved in survival, i.e., growth in cell groups. These SNPs were in the gene *DUF3707*, a prominent gene of the family of pherophorins and genes that regulate *DUF3707* expression, as well as the excretion of pherophorine (Fig. 3). The pherophorine gene family has been suggested to play a major role in the evolution of multicellularity and the expansion of the extracellular matrix in the volvocine algae[43]. Other repeated genomic changes that distinguish an evolved group from evolved single cells in our study are likely involved in mitigating the consequences of growth in cell groups. These include certain metabolism processes and regulative control of cellular processes. These identified changes might facilitate synchronized performance of cells within a group by altering signaling and the corresponding cascades as information signals coming from outside of the cell group might need more enhancer as signals have to pass the extracellular matrix and the cells. Metabolic changes include response to changes in $CO_2$ levels, which might indicate alteration of light intensity within cell groups, and responses in genes involved in nutrient uptake and cytosolic viscosity. Changes in cell communication have also been linked to the synchronization of the cell cyclce[44] and differentiation of cells within groups[45]. Whether the genomic changes we attribute to cell–cell communication would support the transition to cell synchronization and differentiation requires additional experimental tests. Nevertheless, the genomic changes observed in the cell groups suggest that cells of cell groups evolved to compensate some of the consequences of growth in cell groups that affect growth rates.

Other selection pressures for cell groups that could lead to selection of cell groups are environmental stress[46] and more efficient nutrient usage[41,47]. Cell groups and palmelloid formation in *C. reinhardtii* can for example be induced by organic acids[48] or salt stress[49]. We cannot exclude that some of these other selection pressures might have been (temporarily) present in our study, as we also observed the evolution of cell groups in the absence of the predators, although to a significantly lower degree and in only four out of ten *C. reinhardtii* isogenic lines. Several studies have shown that selection on single-celled organisms by predators with gape size limitation can lead to the evolution of cell groups. Another evolutionary outcome in response to predation are post-ingestion defenses, which lower the probability of digestion by the predator. For example, green-algae evolve to grow thicker cell walls[50] or to increase the production of mucilaginous layers surrounding the algal cell[51,52].

We present experimental results showing that the single-celled *C. reinhardtii* evolves growing in cell groups with a stronger convex survival and reproduction trade-off curve when exposed to continuous predation selection compared to selection without the predator present. The evolved cell groups had unique variants involved in keeping cells together after cell division, suggesting a consistent selective response on the genome level. This fairly high degree of repeatability and the small number of generations suggest some degree of determinism for the phenotypic and genomic response in *C. reinhardtii* to predation pressure.

## Methods

**Selection experiment.** We set up a selection experiment using 10 strains from the *Chlamydomonas* culture collection (Table S1) by exposing them to predation. Strains were grown individually in liquid media for 10 days and split into two 50 ml tissue culture flasks containing COOL media[53] with 160 μM of nitrogen. We added a small number of rotifers to one flask per strain (*predation* treatment) and not to the other flask (*control* treatment). We used a facultative asexual clonal line of the rotifer *Brachionus calyciflorus* from stock-cultures (to minimize the possibility for adaptation in the predator population), washed and starved for 24 h before adding

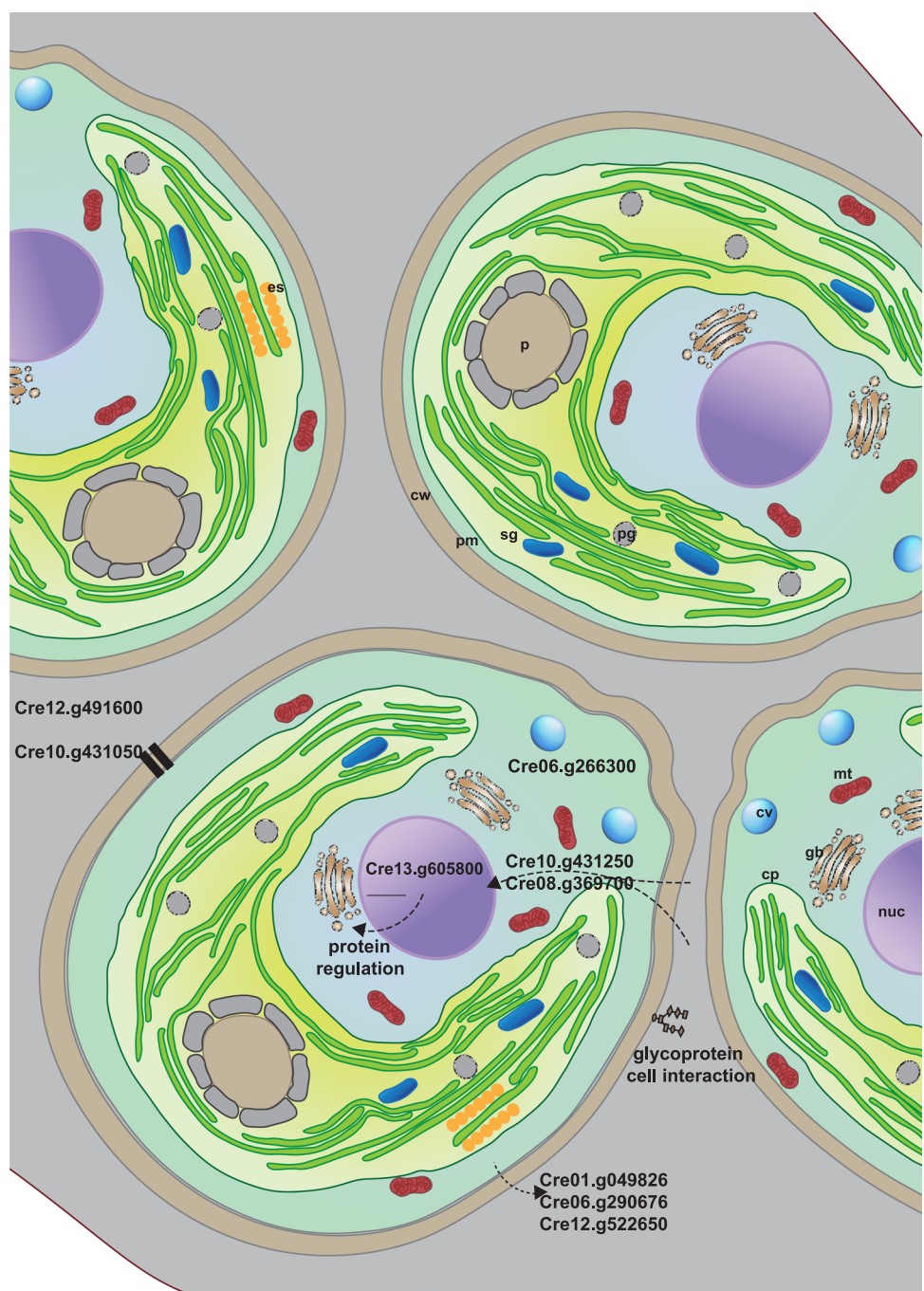

**Fig. 3 Graphical model of genomic changes in cell clusters.** Cells of cell cluster are embedded in an extracellular matrix (gray), which keeps cells together after cell divisions. Processes that are affected by genomic changes observed only in cell clusters are described for the cell at the bottom left. Cell organelles are indicated in the other cells: cw: cell wall, pm: plasma membrane, sg: starch granule, p: pyrenoid, mt: mitochondria, cv: contractile vacuole, cp: chloroplast, gb: golgi body, nuc: nucleus, es: eyespot.

to the selection flasks. The rotifer stock culture is derived from an isolate from Milwaukee harbor, Wisconsin[54] and maintained on *Monoraphidium minutum*. We grew the cultures for six months, replacing ~50% of the culture with fresh medium every 7–10 days without controlling rotifer and algae densities. Cultures were kept at 25 °C in indirect, constant light and static conditions. The number of generations was estimated using a doubling time of 0.4 days (~9 h) and the number of days of the selection experiment. This calculation is based on optimal growth conditions as exact conditions in the selection experiments are not known (changes in nutrient concentrations over time due to consumption by the algal cells and bacteria, as well as recycling of nutrients through rotifer and bacterial excretion).

**Isolation, growth rate, and morphotype classification**. At the end of the experiment, we plated algae on COOL800 ANC 2% agar plates (800 μM nitrogen

containing 100 μg/ml ampicillin, 50 μg/ml neomycin, 75 μg/ml cefotaxime). From these we randomly isolated 12 colonies from the plates for further analyses (hereafter clones). As some colonies did not grow in liquid after isolation for unknown reasons, the number of clones further analyzed differed between strains and selection treatment, with 8–12 clones per strain and selection treatment. Morphology was determined after growing the clones for 3 days in liquid medium (160 μM of nitrogen) under the microscope. Clones were classified into six different morphotypes: non-motile single cells, some motile single cells, majority motile single cells, cell clusters with no single cells, cell clusters with non-motile single cells and cell clusters with motile single cells. Cells were considered motile when they were actively swimming during microscopic observations. In addition, growth rates of evolved clones and the corresponding rotifer growth rates were measured separately. Algal clones were grown with a fixed rotifer population for 3 days in liquid medium (160 μM of nitrogen) and their OD (at a wavelength of 600 nm,

9 sites per well on a Tecan Infinite 200 Pro microplate reader) measured throughout. Then we calculated their growth rates using Malthusian growth parameters[55]. Furthermore, rotifer populations were grown in a similar setup with an evolved algal clone at a fixed initial density of 200,000 cells/ml, rotifers were counted using a stereoscope and their growth rate calculated using Malthusian growth parameters. For the comparison of rotifer and algal growth rates of the isolates we used mixed effect models with strain as random effect and clone morphology (single cells or cell groups) as fixed effects to account for higher relatedness between isolates that were started from the same initial line. We used the package lmerTest[56] in the R environment.

**Sequencing**. To identify potential candidate genes involved in cell groups, we sequenced whole genomes of nine pairs of clones from *predation* and *no-predation* selection experiment. For sequencing, we inoculated 75 ml of COOL800 medium from a single colony of each isolate and let the algae grew under standardized and constant conditions for 10 days (25 °C in indirect, constant light with shaking). Algae were harvested by centrifugation and pellets were transferred to −80 °C before DNA extraction. DNA was extracted using CTAB-based method without cell disruption and treated with RNase from Qiagen®[57]. We prepared the libraries for sequencing (2x150bp) using the Nextera DNA Library Preparation Kit from Illumina® and sequenced using the NextSeq Illumina platform.

**Sequence data analyses**. Quality filtering (cutoff of Q20) and adaptor trimming of raw data was performed in CLC Genomicsworkbench V9 (Qiagen Bioinformatics), and any reads <100 bp were discarded. All orphan (not paired) reads were also discarded. The successfully paired reads (PE) were retained after quality control and duplicate elimination (~1%) and processed further. First, PE data were assembled using the assembly pipeline in CLC Genomicsworkbench using standard parameters (wordsize = automatic (ranged between 21 and 23), bubble size = automatic (estimated = 50)) and results are summarized in Table S2.

For the variant analyses, the resulting assemblies have been local realigned using the read track mode. As a reference, we used the *C. reinhardtii*_281 genome V5.0 with its annotations V5.5. Variant analyses were done within the resequencing analyses of CLC using the fixed ploidy and a sequence error model with parameters estimated from the data. Settings have been required variant analyses probability 90% minimum coverage of 10 and a minimum frequency of 20% with a significance of 1.0. Analyzing the fixed variants unique for the control versus grazed strains was conducted by first grouping variant-read-tracks files of the grazed experiments (S1, S3, S5, S7, S9, S11, S13, S15, S17) as well as those for the control set (S2, S4, S6, S8, S10, S12, S14, S6, S18), and unique variants of the grazed and control genomes have been extracted. The unique set of variants were annotated by filter based on overlaps against the *C. reinhardtii* annotation files V5.5. Functional consequences were extracted under resequencing analyses, and the functional consequences option for amino-acid changes in the CLC work package and functional annotations were added using phytozyme[58].

**Chemostat experiment and transcriptome analysis**. We picked one-pair of *cell group* and *single cell* isolate and inoculated them to 18 chemostats in COOL80 medium. After 10 days we either added rotifers (rotifer treatment), switched the resource supply to lower nitrogen concentration (low-nitrogen treatment; 40 μM nitrogen) or kept the algae as they were (control treatment), each treatment was replicated three times. After 5 additional days we harvested the algae from the chemostats. We compared the algae from the chemostats by measuring cell group sizes (number of cells per group), percentage of cell group, and overall population sizes for the three treatments (Fig. S3). Simultaneously, the harvested algae were prepared for total RNA extraction with RNeasy Plant Mini Kit from Qiagen® for RNA-Seq analysis. We pooled the RNA extracted by treatment and RNA-sequenced it using Illumina HiSeq (2x150bp). We used isolates from the selection lines that were started with cc1009 ("wildtype") and an isolate growing as cell groups with no single cells from the predation line, and a single cell isolate from the non-predation line.

We mapped the high-quality reads identified by FastQC[59], to *Chlamydomonas* version 5.0 reference genome[33] using hisat2 software[60], on average 9.7 million reads were mapped per sample. For samples from rotifer predation treatment there was potential contamination, identified by the relative higher percentage of reads that did not map to the *Chlamydomonas* genome (~40%), the other samples mapped similarly (~75%), while only a minority of reads mapped more than two times to the reference genome. Our gene list was comprised of 16,707 genes, we used HTSeq[61] to count the reads per gene, and DESeq2[62] to normalize the read counts of every sample by estimation of size factors and estimation of dispersion, which normalized the depth and dispersion parameters of every sample so to make the samples comparable (Tables S1 and S2). We used the normalized read counts to verify the correlations between our strain-environment combinations, and to create a multi-dimensional scaling (MDS) analysis based on Euclidean distances. We used Spearman correlation method because it better preserves the relative rank relationships between samples and is less influenced by skewness and outliers. We compared samples based on clones and treatments by estimating the log fold change of every gene and the corresponding *P*-value adjusted for multiple comparisons.

## Data availability

The RNA-seq data generated in this study have been deposited in the GEO database under accession code GSE166685. The raw and normalized read count tables regarding the RNA-seq data are provided as Supplementary Information in the GEO database under accession code GSE166685. Source data are provided as a Source Data file (Figs. 1, 2, S2, S3). The DNA sequence data generated in this study have been deposited in the NCBI database under the accession code GSE166685. Source data are provided with this paper.

## Code availability

Github repository https://doi.org/10.5281/zenodo.4773003. (https://github.com/jpimentabernardes/Transcriptome_Trade-offs).

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

## Acknowledgements

We thank Julia Haafke for help with the selection and transcriptome experiment and the preparations for sequencing, Sven Künzel and his team for help with sequencing the genomes, the Cologne Genome Center for help with sequencing the transcriptomes. We thank Santosh Sathe for comments on the manuscript. This work was funded by the German Research Foundation (DFG; grant BE 4135/4-1 and BE 4135/4-2 to L.B.) and the Max Planck Society (to L.B.).

## Author contributions

L.B. conceived the study. N.W. and L.B. designed the selection study, M.V. and L.B. designed the transcriptome study, U.J. and L.B. designed the genome study. N.W. performed the selection experiment and data collection, M.V. performed transcription experiment, M.V. and J.B. performed transcriptome analyses, U.J. ran genome analyses, R.H. ran experiment for Fig S1. All authors discussed the results. All authors contributed to the writing of the manuscript.

## Funding

## Competing interests

The authors declare no competing interests.
