## [Transparent Peer Review File · Nature Communications]

REVIEWER COMMENTS

Reviewer #1 (Remarks to the Author):

The evolution of convex trade-offs enables the transition towards multicellularity

General

The authors investigated ecological trade-offs in single and groups of *Chlamydomonas* cells in response to predation. They identified a convex trade-off curve between survival and reproduction in the evolution of cell groups. Interestingly, the trait was heritable and, surprisingly, the same mutations emerged in different groups. 500 generations of predator selection resulted in a convex trade-off between survival and reproduction, with a heritable genotype-phenotype map.

This is certainly an interesting manuscript and the authors have thought through most of the experimental issues well (except perhaps the nitrogen deplete experiments). The results are clear and, refreshingly, the authors do not overstate their interpretation. The work is mostly very nice to read. There are some issues that, in my view, are important to address. I detail the comments I have below, roughly as they arise in the manuscript.

Comments

1. Figure 1 lays out the conceptual framework as well as predictive hypothesis for the paper. It should be better referenced. It appears to be a re-do of Figure 3 from reference 19 or a summary of Figure 3 from reference 30. The basic mathematical model was developed in reference 25.
2. The authors have adopted the oversimplified and artificially distinct alternatives of staying together or coming together: “Previous work has described at least two mechanisms leading to multicellularity evolution: (1) aggregation of genetically diverged free-living cells like in cellular slime molds and *Myxobacteria* or (2) cellular adhesion of clones unable to fragment after cell division like in the *Chlamydomonas* green algae or in *Saccharomyces* yeasts.” This is only a heuristic and I think it would be fair for the authors to say so. In the real world (and in *Chlamydomonas*) there is nothing precluding a mix of these two mechanisms. As Okasha (book, 2007, chapters 6 to 8) and Durand (book, 2020, chapter 16) infer, the distinction should not be a category one. Certainly, in response to predation in *Chlamydomonas* and *Chlorella*, cells ‘stay together’ (Kapsetaki, Fisher, West 2018 etc) but they also ‘come together’ (Sathe&Durand 2015; Kapsetaki, Fisher, West 2018 etc). This was actually one of the express aims of Kapsetaki’s work. She and colleagues quite clearly show that predators induce a mix of the two mechanisms in *Chlamydomonas* and other green algae. This has tremendous significance for experiments such as these, so while I think it is fine to work with the heuristic for experimental convenience, it would be nice for the reader to be aware of this.
3. The authors say that “we found a significant lower frequency of swimming cells in isolates growing in groups” Lines 34-35. This confuses me slightly, because the data in Fig 1B suggest more groups were formed when cells were motile (subcategory 3). In fact increased motility is reportedly more associated

with group formation by cells coming together so perhaps the authors quote above pertains only to groups that are already established as a stable phenotype and once established the cells deflagellated and lose motility? I think this is the case from my reading of the paper.

4. Related to point 1, I would draw the authors' attention to the purely philosophical argument concerning causal inferences (e.g., Okasha, 2007), made many times now, that genetically homogeneous groups are not a prerequisite for multicellularity. To the authors' credit they do say later that: "Thus, trade-off curves are most informative when comparing estimates from intraspecific clones." Experimentally, and for the aims of this submission it is indeed more informative to compare estimates from intraspecific clones, but to remain true to the philosophical works, it may be helpful to say that in the evolutionary history of multicellularity, clonality is a product. Kin selection leads to the close genetic relationships, but that does not mean that a priori the close genetic relationships were a requirement (Okasha, book, 2007, and see the 'cart-before-the-horse' argument in Durand, book, 2020). In fact, this is one of the reasons the experiments such as the ones in this submission are so helpful. Trade-offs may be just as important, if not more important than genetic relationships in the division of labor that the authors refer to (e.g., Michod's covariance effect demonstrates this). If the trade-offs were so important, the findings in this submission are helpful to understand all instances of predation-induced groups Those comprising genetically identical groups and those comprising different strains or even species (see references above).

5. Line 52-53: "...through lower uptake OF resources". There are several other syntactical or grammatical errors that I haven't listed.

6. In Fig 1d. Is there an explanation why the rotifer growth rate is lower in non-motile cells? The expectation was a greater growth rate (cells can't swim away).

7. Lines 73-78. I do think that while the authors speak about concave and convex curves, they need to explicitly mention the 'covariance effect' (the reference to Michod's works), because this is such an important driver in the division of labor argument that they refer to.

8. Perhaps for greater clarity. Line 83: the authors refer to isogenic lines, and then say they do not confine selection to one genotype (line 86). Are you referring to the different strains in table S3 and mean that each strain is an isogenic line? I think so, but the way it is written is initially confusing.

9. The statement (line 111-112): "We found significantly more cell groups that form through clonal reproduction.....". How could it not be clonal, didn't you start with clones at the beginning of each experiment? Or are you referring to the 'coming together' part of aggregation?

10. Can the authors say why 4 out of 10 lines formed groups even if there weren't predators? Is it chance / drift? They provide some references (6,27 and 28) but can they suggest why the controls formed groups. Groups can form by the prey sensing predators in conditioned medium even when they no longer there (Sathe and Durand 2015), but I m assuming there was no chance of that in the experimental setup used in this submission. Other possible explanations may be contamination by bacteria, a change in nutrients, temperature flux. Perhaps the authors can speculate?

11. The finding that non-motile isolates had a lower growth rate for similar survival rates, is that because of sinking (I note the cultures were static—no stirring or shaking) and relative light deprivation?

12. Line 248. In addition to ref 18, I think that reference to Kapsetaki's work is also warranted. Although they documented the 'coming together' later than in ref 18, they highlighted the mix of staying together and coming together in group formation.

13. The motivation for also using N deplete conditions is not clear. The authors introduce this for the first time under gene expression patterns in the results. They say this is to mimic low resource conditions, but the thinking around this is not explained.

14. Is it said somewhere what the nitrogen deplete conditions were? Was there no (zero) nitrogen or was it just a lower concentration?

15. Interesting the differentially expressed genes show very little change between predation and N starvation (Fig 2c). As the authors suggest, and I would agree, the changes in gene expression in metabolism and cell cycling that were observed are due to cell stress. The explanation for this surely lies in the association between cell stress and either nutrient depletion like N starvation (Sathe et al, 2019, Phycological Research) and / or group formation (Bouderbala et al Acta Biotheoretica 2018, Sathe&Durand 2015). Programmed cell death is very much a feature of nitrogen depletion in Chlamydomonas cultures, but it is noted that the authors don't report cell death in either the predator-induced groups or in the N deplete groups. Perhaps the stress from N starvation wasn't strong enough to induce death (as I say in point 13 it is not clear if there was no nitrogen or just a lower concentration), but certainly the change in metabolism and cell cycling supports the notion that there is cell stress and decreased cell division.

16. The authors say: "While cell groups in the present study are the result of growth in groups where the daughter cells fail to separate after cell division (Fig. S2),...." It is not clear what they mean by separate, and FigS2 does not show this. I think this is important. Are the cells simply sticking together by ECM or is there a failure of the cell septum to form? This helps describe the very first step towards coloniality, see the next point. From what I can tell, it seems to me that the cells may simply be sticking together. In other words, it is not the case that there is failure of septum formation. Either way it is not clear from this manuscript what the case is.

17. The authors say that "Previous work using comparative genomics suggests that changes in genes involved in cell cycle regulation and synchronization are associated with the initial step of growing in groups³⁶". This is true but not for the reason they cite. The context is misleading. The reference (36) to the comparative genomics using *Gonium* was not the "initial" step as the authors claim. The initial step, based on morphology and ultrastructure, seems to be failure of cells to cleave (see 2013 Arakaki et al's work on the ultrastructure of *Tetrabaena socialis*). The authors also need to look at the comparative genomics of this initial step in *T socialis* (Featherston et al, Mol Biol Evol, 2018) I think they referring to this organism if they talking about the initial step. I do realize this has been confusing because the ultrastructural finding of cellular bridges in *Tetrabaena* wasn't expected based on previous works—but

the cellular bridges in *T. socialis* certainly exist as reported by Arakaki et al, 2013 (The Simplest Integrated Multicellular Organism Unveiled) and the changes in cell cycle gene number were also found in *T. socialis*.

Reviewer #2 (Remarks to the Author):

This manuscript reports the results of an in vitro evolution experiment involving an (initially) single-celled alga and a rotifer predator. Multi-celled groups evolved in 10 of 10 microcosms including predator and prey and 4 of 10 including only the algae. A trade-off between survival and fecundity evolved to be convex in the multi-celled groups, a situation predicted to facilitate the evolution of a somatic/reproductive division of labor.

The manuscript is clear and succinct, and the measurement of survival/fecundity trade-off curves in genetically diverse, experimentally evolved populations is a significant accomplishment with important implications for the evolution of cellular differentiation. This is a nice experimental test of an influential theoretical framework that has previously only been tested using comparative methods. I have suggested some minor changes and clarifications below, but overall I think this is solid and important work appropriate for publication in *Nature Communications*.

In the interest of full disclosure, I'll mention that Rick Michod was my PhD co-advisor, and this work constitutes an experimental test of one of his main theoretical results. I do not feel that this has prevented me from impartially evaluating the manuscript.

Minor comments:

Lürling & Beekman (2006) is cited in support of cellular adhesion as a mechanism of multicellularity evolution, but my understanding is that Lürling and Beekman report phenotypically plastic, not evolutionary, change. Other papers, including some by the corresponding author, could be cited as more appropriate examples.

71-73: the description here ("when an increase in survival comes at a cost of reduced reproduction (Fig. 1a, solid line; convex trade-off curve)") doesn't match the figure. Survival comes at a cost of reduced reproduction in both the convex and the concave curves.

116-118: rotifer growth rate is an indirect way of estimating algal survival, but I suppose it works.

133: the enlargement constraint has not previously been described. Self-shading and reduced nutrient uptake per unit volume due to decreased surface area to volume ratio are other possible costs (or maybe this is what is meant by enlargement constraint?).

135-136: Figure 1D appears to show that trade-off curves differ between motile and non-motile isolates. I don't see that it shows the frequencies of swimming cells, so I don't know how it supports the difference in frequency this sentence attributes. The chi-square test does support this, but it would be nice to know what the frequencies were.

149-153: I find this section confusing. It's not clear to me how lower correlation scores imply less variable transcription profiles (I would have thought the opposite). It's also not clear what comparisons are being made here. The claim is that correlation scores are lower in cell groups than in single cells, but in both examples only one correlation score is given (so how is this a comparison?). The scores reported seem to be correlations between single cells and cell groups (lower left of the figure); the relevant

scores for the claim would seem to be a comparison between single-cell correlations (upper left) and group correlations (lower right).

174-176: I would be interested to know how many of these variants evolved during the experiment, which could be estimated either by comparing isolates of the same starting strain across treatments (any variant found in both was probably present in the ancestor) or by comparison to pre-existing WGS data. I don't seem to have access to Table S3, so I don't know what strains were used in the experiment, but many of the Chlamy Center strains have been sequenced by Jang et al. (<https://doi.org/10.1371/journal.pone.0041307>) and by Craig et al. (<https://doi.org/10.1111/mec.15193>).

183-197: it might be useful to compare genetic variants, gene expression differences, and functional assignments of both to a previous genetic analysis of experimentally evolved multi-celled groups of *C. reinhardtii* (<http://dx.doi.org/10.1098/rsos.180912>).

229: volvocine is not a formal taxon and does not need to be capitalized.

231: "aggregates" is often contrasted with clonally developing cell groups (such as *Gonium* and *Pandorina*), so it's probably not a good choice to describe them.

233: the abbreviation ECM is only used a couple of times and is probably unnecessary.

242: reference 35 does not report evolution of cell groups in response to predation.

247-248: I would say "in some other studies", since this is not true of refs 8, 9, 12, and 35.

258: reference 12 does not report gene expression studies (genetics of that experiment are reported in <http://dx.doi.org/10.1098/rsos.180912>).

262: in the discussion of Duf3707, it may be worth mentioning that expansions in the pherophorin gene family are one of only a handful of large-scale differences between the *Chlamydomonas* and *Volvox* genomes, and these genes have been suggested to have played important roles in the evolution of multicellularity and the expansion of extracellular matrix in the volvocine algae.

299-300: I'm not sure what message the authors intend with "the existing genomic and phenotypic repertoire of the cells has been used to respond to selection." This seems self-evident and trivial, which is probably not the intent for the final sentence of the discussion. Don't all responses to selection use the existing genomic and phenotypic repertoire of the organisms under selection? I suspect that the authors can draw a stronger conclusion from the striking repeatability of their experimental populations.

307-308: the provenance of the rotifers is vague. Where did the stock cultures originate? How were they confirmed to be *B. calyciflorus*?

309-310: on what were the rotifers fed before starvation? Is 24H enough, or is it possible that some of the detected contamination was carried over food from before the experiment?

The length of the experiment is reported as six months and as ~500 generations, but it's not clear to me how the generation time was estimated.

In the supplemental spreadsheets, I'm not sure what homozygous and heterozygous mean in the context of a haploid organism.

Matt Herron

Reviewer #3 (Remarks to the Author):

In the manuscript 'The evolution of convex trade-offs enables the transition towards multicellularity' Bernades et al report the results from an experimental evolution study, where the green algae

Chlamydomonas reinhardtii and the rotifer *Brachionus calyciflorus* were co-evolved. The authors report the evolution of a convex trade-off between reproduction and survival for cell groups, which has previously been predicted to facilitate the differentiation into soma and germ-like cells. Further, gene expression analysis and whole genome sequencing is employed for a subset of evolved isolates to compare gene expression patterns and genomic changes between single cells and cell groups.

The topic is timely and I very much like the approach of the study. Overall, in particular the methods part lacks significant detail that hinders replicating this study. I have several comments/ questions:

1) I24-26/I73-76: The transition to multicellularity does not necessarily follow the route described here – the differentiation in soma and germ-like cells does only apply for a subset of the cases of clonal multicellularity. Whereas for mammals this is true, it is not so for sponges, for example.

2) I36: I am not convinced that the convex trade-off evolved in cell groups as claimed here as the authors do not present the data from the ancestral lines. Also, the convex trade-off is also in the evolved single cells. It might be a general feature of the model system in that particular environment, i.e. with predators. From the Figure legend it is not clear whether the data presented is only from the predation treatment or also includes the no predation. It would have been interesting to compare the trade off between treatments.

3) It is interesting that the authors differentiate between soma and germ line cells dependent on whether cells are part of the cell group or not. The question is whether cell groups can be assigned to be soma. If this was the case, somatic cell groups would be unable to give rise to new cell-groups – per definition they would be an evolutionary dead end. The authors do not show any data to support that cell groups actually can be regarded as soma. A study by Herron et al (2019), which also used the very same prey and predator species in an experimental evolution study, report a variety of cell groups with multicellular life cycles, which themselves give rise to new cell groups. This contradicts the assignment of cell groups as soma.

4) *C. reinhardtii* is able to form palmelloids – how do the authors differentiate between them and the evolved cell groups? Herron et al (2019) observed palmelloids among their evolved groups that originated from cells staying together (clonal origin) versus from coming together (aggregative). They also evolved a heritable clonal group morphology, so that Bernandes et al cannot claim “response to predation has been observed in other studies as a phenotypically induced responses^{6,18} rather than a heritable phenotype. Differences the observation in our and other study are likely the result of the underlying process of how the cell groups form. While cell groups in the present study are the result of growth in groups where the daughter cells fail to separate after cell division (Fig. S2), in other studies cells actively aggregated for the formation of the group of cells¹⁸.”

5) Measurement of algal growth rates as OD: I worry about reliably measuring growth rates as ODs as done here as some of the isolates are a mixture of cell groups, single cells, and predators. Also, how can one measure growth of cell groups as OD? I think that estimates of cell and group densities would need to happen as done here for the predators via counting or via image analysis as done by Herron et al (2019). I particularly worry about measuring growth of groups but also that of cells. For cells, the OD would only provide a good estimate of growth that is comparable among the different isolates if cell size among the different isolates is the same.

6) *C. reinhardtii* faces the constraint between cell division and motility. How does this relate to the different groups with motile/non-motile cells as observed here? In particular, how can the authors exclude that they were observing different parts of the cell/ group life cycle as for example in Ratcliff et al 2013 and described by Herron et al 2018: “...milestones in the life cycle: at 3 h, clusters appear

dormant; at 6 h, unicellular propagules are actively swimming; at 9 and 12 h, many have lost motility and begun to develop into multicellular clusters; and at 48 h, clusters have reached a large size (approx. 50 cells or more).“

7) 12 colonies were randomly picked at the end of the experiment from each strain/predator combination:

- How are these representative of the composition in each population? How closely related are they?

Here I worry about treating each isolate as an independent replicate, e.g. in Figure 1c, d. How do you argue against pseudo-replication? To me your experimental set up results in 10 replicates per predation/no-predation treatment. Without knowledge of the underlying relatedness within one flask, one needs to account for the particular strain/predator combination in all analyses.

- Can cells and cell-groups be distinguished due to colony morphology? If so how was the randomness of the colony choice regarding predator treatment ensured?

8) Sequencing: It is not clear which pairs of clones were chosen for sequencing. It is stated: I333-335: “To identify potential candidate genes involved in cell groups, we sequenced whole genomes of nine pairs of clones from predation and no-predation selection experiment.” I would have expected 10 pairs of clones to be sequenced if there was one representative clone chosen per strain x predation combination. Why are there only nine? Also, I would have liked information on the colony morphology send for sequencing – the reader can only assume that cell groups were chosen from the predation treatment and single cells from the predation-free treatment. Nevertheless, when looking Figure 1b, the majority of isolates comprise a mixture of cell groups and single cells. There is no information on the amount of time the colonies were grown before sending to sequencing. Were the cultures checked for morphological heterogeneity, i.e. were there mixed cultures, before DNA extraction?

9) Chemostat experiment and Transcriptome analysis: One pair of cell group and single cell isolate was chosen (I366/367). I would have liked to know why this particular pair was chosen. Did this stem from the same strain/predation combination, i.e. the same flask? What was the cell-group morphology? For the growth in the chemostats, to me it is not clear why an additional treatment, i.e. low nitrogen, was included. Also, the number of replicates do not add up – I am correct that 18 chemostats (and not 9) were used?

Overall, the manuscript needs careful checking in terms of language and grammar, e.g. I51: “preyed upon” instead “fed on”; I52: “of” instead “for”; I129: “having” instead “have”, I367: “chemostats” instead “chemostat”

Other comments:

I. 32: which trait? – cell groups?

Figure S1, I10: remove: “(see STAR methods)

Reviewer comments, *our response*, and alterations to the text.

Reviewer #1 (Remarks to the Author):

The evolution of convex trade-offs enables the transition towards multicellularity

General

The authors investigated ecological trade-offs in single and groups of *Chlamydomonas* cells in response to predation. They identified a convex trade-off curve between survival and reproduction in the evolution of cell groups. Interestingly, the trait was heritable and, surprisingly, the same mutations emerged in different groups. 500 generations of predator selection resulted in a convex trade-off between survival and reproduction, with a heritable genotype-phenotype map.

This is certainly an interesting manuscript and the authors have thought through most of the experimental issues well (except perhaps the nitrogen deplete experiments). The results are clear and, refreshingly, the authors do not overstate their interpretation. The work is mostly very nice to read.

Reply: We thank the reviewer for the positive evaluation. We hope we could clarify all points raised by the reviewer.

There are some issues that, in my view, are important to address. I detail the comments I have below, roughly as they arise in the manuscript.

Reply: Please find our replies in italic below.

Comments

1. Figure 1 lays out the conceptual framework as well as predictive hypothesis for the paper. It should be better referenced. It appears to be a re-do of Figure 3 from reference 19 or a summary of Figure 3 from reference 30. The basic mathematical model was developed in reference 25.

Reply: We added the reference to the legend.

2. The authors have adopted the oversimplified and artificially distinct alternatives of staying together or coming together: “*Previous work has described at least two mechanisms leading to multicellularity evolution: (1) aggregation of genetically diverged free-living cells like in cellular slime molds and Myxobacteria or (2) cellular adhesion of clones unable to fragment after cell division like in the Chlamydomonas green algae or in Saccharomyces yeasts.*” This is only a heuristic and I think it would be fair for the authors to say so. In the real world (and in *Chlamydomonas*) there is nothing precluding a mix of these two mechanisms. As Okasha (book, 2007, chapters 6 to 8) and Durand (book, 2020, chapter 16) infer, the distinction should not be a category one. Certainly, in response to predation in *Chlamydomonas* and *Chlorella*, cells ‘stay together’ (Kapsetaki, Fisher, West 2018 etc) but they also ‘come together’ (Sathe&Durand 2015; Kapsetaki, Fisher, West 2018 etc).

This was actually one of the express aims of Kapsetaki’s work. She and colleagues quite clearly show that predators induce a mix of the two mechanisms in *Chlamydomonas* and other green algae. This has tremendous significance for experiments such as these, so while I

think it is fine to work with the heuristic for experimental convenience, it would be nice for the reader to be aware of this.

Reply: The reviewer is correct, and we added that it is also possible that there is a mix of the two mechanisms. L44: The initial evolutionary step involves the growth of undifferentiated cells in groups (hereafter: cell groups; for a review see e.g.,²). Previous work has described at least two mechanisms, as well as a combination, leading to the evolution of multicellularity ...

3. The authors say that “we found a significant lower frequency of swimming cells in isolates growing in groups” Lines 34-35. This confuses me slightly, because the data in Fig 1B suggest more groups were formed when cells were motile (subcategory 3). In fact increased motility is reportedly more associated with group formation by cells coming together so perhaps the authors quote above pertains only to groups that are already established as a stable phenotype and once established the cells deflagellated and lose motility? I think this is the case from my reading of the paper.

Reply: The significance statement refers to isolates from the predation and no-predation lines. We say now in L144: Indeed, we found a significant lower frequency of swimming cells across all isolates in isolates growing in groups ($\chi^2= 6.85$, $df=1$, $p=0.009$; Fig. 1b).

4. Related to point 1, I would draw the authors’ attention to the purely philosophical argument concerning causal inferences (e.g., Okasha, 2007), made many times now, that genetically homogeneous groups are not a prerequisite for multicellularity. To the authors’ credit they do say later that: “Thus, trade-off curves are most informative when comparing estimates from intraspecific clones.” Experimentally, and for the aims of this submission it is indeed more informative to compare estimates from intraspecific clones, but to remain true to the philosophical works, it may be helpful to say that in the evolutionary history of multicellularity, clonality is a product. Kin selection leads to the close genetic relationships, but that does not mean that a priori the close genetic relationships were a requirement (Okasha, book, 2007, and see the ‘cart-before-the-horse’ argument in Durand, book, 2020). In fact, this is one of the reasons the experiments such as the ones in this submission are so helpful. Trade-offs may be just as important, if not more important than genetic relationships in the division of labor that the authors refer to (e.g., Michod’s covariance effect demonstrates this). If the trade-offs were so important, the findings in this submission are helpful to understand all instances of predation-induced groups Those comprising genetically identical groups and those comprising different strains or even species (see references above).

Reply: We thank the reviewer for pointing us towards the philosophical discussion. We added to the introduction (L84-85): Thus, trade-off curves are most informative when comparing estimates from intraspecific clones although clonality is in general not a requirement for the evolution of multicellularity²⁹.

5. Line 52-53: “...through lower uptake OF resources”. There are several other syntactical or grammatical errors that I haven’t listed.

Reply: We corrected this and other mistakes.

6. In Fig 1d. Is there an explanation why the rotifer growth rate is lower in non-motile cells? The expectation was a greater growth rate (cells can't swim away).

Reply: In an independent study, we measured the rotifers' ingestion rates for different Chlamydomonas isolates and densities and found that motility/non-motility does not affect ingestion rates, likely because the current for feeding that the rotifer creates is too strong for the algal cells to escape. As this other study is work in progress, we cannot refer to this.

7. Lines 73-78. I do think that while the authors speak about concave and convex curves, they need to explicitly mention the 'covariance effect' (the reference to Michod's works), because this is such an important driver in the division of labor argument that they refer to.

Reply: We added the covariance effect to our description. We say now (L74-80): The second step in the evolutionary transition toward multicellularity, the soma-germ separation, is more likely expected when the trade-off shifts towards convex curvature with increasing group size and when the fitness trade-off creates a covariance effect²⁰. The covariance effect at the group level allows a higher group fitness compared to the average fitness of the component cells (soma and germ) and could thus lead to a situation where the benefits eventually outweigh the reproductive costs of cell groups.

8. Perhaps for greater clarity. Line 83: the authors refer to isogenic lines, and then say they do not confine selection to one genotype (line 86). Are you referring to the different strains in table S3 and mean that each strain is an isogenic line? I think so, but the way it is written is initially confusing.

Reply: We changed the wording to (L85-87): Here, we made use of experimental evolution to evolve intraspecific trait diversity in ten cell lines representing distinct genotypes of the unicellular green algae Chlamydomonas reinhardtii.

9. The statement (line 111-112): "We found significantly more cell groups that form through clonal reproduction.....". How could it not be clonal, didn't you start with clones at the beginning of each experiment? Or are you referring to the 'coming together' part of aggregation?

Reply: Yes, we used this phrase to refer to the clonal growth within a cell group. In order to avoid misunderstanding we changed the wording to (L114-115): We found significantly more cell groups where cells adhered after cell division to each other

10. Can the authors say why 4 out of 10 lines formed groups even if there weren't predators? Is it chance / drift? They provide some references (6,27 and 28) but can they suggest why the controls formed groups. Groups can form by the prey sensing predators in conditioned medium even when they no longer there (Sathe and Durand 2015), but I m assuming there was no chance of that in the experimental setup used in this submission. Other possible explanations may be contamination by bacteria, a change in nutrients, temperature flux. Perhaps the authors can speculate?

Reply: Good point, we added the following to the discussion (L328-330): Cell groups and palmelloid formation in C. reinhardtii can for example be induced by organic acids⁴⁸

or salt stress⁴⁹. We cannot exclude that some of these other selection pressures might have been (temporarily) present in our study, as we also observed the evolution of cell groups in the absence of the predators, although to a significantly lower degree and in only four out of ten *C. reinhardtii* isogenic lines.

11. The finding that non-motile isolates had a lower growth rate for similar survival rates, is that because of sinking (I note the cultures were static—no stirring or shaking) and relative light deprivation?

Reply: We don't have data for this interpretation. We added a list of possibilities to the discussion: (L260-263): The observed lower growth rate for a given survival rate of non-motile cells compared to motile cells and selection in static cultures in our experiments suggest that motility could be beneficial for accessing nutrients in the planktonic phase and/or access to light.

12. Line 248. In addition to ref 18, I think that reference to Kapsetaki's work is also warranted. Although they documented the 'coming together' later than in ref 18, they highlighted the mix of staying together and coming together in group formation.

Reply: Thanks for this suggestion, we added the reference.

13. The motivation for also using N deplete conditions is not clear. The authors introduce this for the first time under gene expression patterns in the results. They say this is to mimic low resource conditions, but the thinking around this is not explained.

Reply: The reviewer is correct. We used the N deplete conditions to mimic low resource competition, an environment the cells experienced during the selection experiment. We added to the text now (L156-159): with predation by rotifers (+R), with reduced nitrogen concentration to resemble low resource conditions (-N; we halved the concentration of N to 40 μ M) or without any of the two stressors that the populations experienced in the selection experiment (control C).

14. Is it said somewhere what the nitrogen deplete conditions were? Was there no (zero) nitrogen or was it just a lower concentration?

Reply: The N concentration was reduced to 40 μ M. We say this in the Methods section describing the chemostat experiment. However, to make this point more prominent as we now added to the main text (L157-158): we halved the concentration of N to 40 μ M.

15. Interesting the differentially expressed genes show very little change between predation and N starvation (Fig 2c). As the authors suggest, and I would agree, the changes in gene expression in metabolism and cell cycling that were observed are due to cell stress. The explanation for this surely lies in the association between cell stress and either nutrient depletion like N starvation (Sathe et al, 2019, Phycological Research) and / or group formation (Bouderbala et al Acta Biotheoretica 2018, Sathe&Durand 2015). Programmed cell death is very much a feature of nitrogen depletion in *Chlamydomonas* cultures, but it is noted that the authors don't report cell death in either the predator-induced groups or in the N deplete groups. Perhaps the stress from N starvation wasn't strong enough to induce death (as I say in point 13 it is not clear if there was no nitrogen or just a lower concentration), but

certainly the change in metabolism and cell cycling supports the notion that there is cell stress and decreased cell division.

Reply: The reviewer is correct that we would expect programmed cell death under N-starvation. As we wanted to test for stronger N-limitation and thus slower growth, we did not switch to N-free medium but only a lower N concentration. See to our two replies above, appreciate the comments and added to the main text (L157-158): we halved the concentration of N to 40µM.

16. The authors say: “While cell groups in the present study are the result of growth in groups where the daughter cells fail to separate after cell division (Fig. S2),...” It is not clear what they mean by separate, and FigS2 does not show this. I think this is important. Are the cells simply sticking together by ECM or is there a failure of the cell septum to form? This helps describe the very first step towards coloniality, see the next point. From what I can tell, it seems to me that the cells may simply be sticking together. In other words, it is not the case that there is failure of septum formation. Either way it is not clear from this manuscript what the case is.

Reply: This is a good point and needs clarification in the MS. We don't know if cells are sticking together by the ECM or a failure of the cell septum to form. We added this to the text (L272-275): It is possible that cells were kept together by the extracellular matrix or failure of the cell septum to form. The observed genomic changes suggest the former (see below and Fig. 3) but further analyses are required to identify the underlying mechanism.

17. The authors say that “Previous work using comparative genomics suggests that changes in genes involved in cell cycle regulation and synchronization are associated with the initial step of growing in groups³⁶”. This is true but not for the reason they cite. The context is misleading. The reference (36) to the comparative genomics using *Gonium* was not the “initial” step as the authors claim. The initial step, based on morphology and ultrastructure, seems to be failure of cells to cleave (see 2013 Arakaki et al’s work on the ultrastructure of *Tetrabaena socialis*). The authors also need to look at the comparative genomics of this initial step in *T socialis* (Featherston et al, *Mol Biol Evol*, 2018) I think they referring to this organism if they talking about the initial step. I do realize this has been confusing because the ultrastructural finding of cellular bridges in *Tetrabaena* wasn’t expected based on previous works—but the cellular bridges in *T socialis* certainly exist as reported by Arakaki et al, 2013 (*The Simplest Integrated Multicellular Organism Unveiled*) and the changes in cell cycle gene number were also found in *T socialis*.

Reply: We thank the reviewer for pointing to this. We slightly modified the sentences by adding a word to avoid a misleading interpretation (L298-300): Previous work using comparative genomics suggests that changes in genes involved in cell cycle regulation and synchronization are associated with the evolution of multicellularity⁴⁰.

Reviewer #2 (Remarks to the Author):

This manuscript reports the results of an in vitro evolution experiment involving an (initially) single-celled alga and a rotifer predator. Multi-celled groups evolved in 10 of 10 microcosms including predator and prey and 4 of 10 including only the algae. A trade-off between survival and fecundity evolved to be convex in the multi-celled groups, a situation predicted to facilitate the evolution of a somatic/reproductive division of labor.

The manuscript is clear and succinct, and the measurement of survival/fecundity trade-off curves in genetically diverse, experimentally evolved populations is a significant accomplishment with important implications for the evolution of cellular differentiation. This is a nice experimental test of an influential theoretical framework that has previously only been tested using comparative methods. I have suggested some minor changes and clarifications below, but overall I think this is solid and important work appropriate for publication in Nature Communications.

Reply: We thank the reviewer for the positive evaluation. We hope we could clarify all points raised by the reviewer.

In the interest of full disclosure, I'll mention that Rick Michod was my PhD co-advisor, and this work constitutes an experimental test of one of his main theoretical results. I do not feel that this has prevented me from impartially evaluating the manuscript.

Minor comments:

Lürling & Beekman (2006) is cited in support of cellular adhesion as a mechanism of multicellularity evolution, but my understanding is that Lürling and Beekman report phenotypically plastic, not evolutionary, change. Other papers, including some by the corresponding author, could be cited as more appropriate examples.

Reply: We changed the references.

71-73: the description here ("when an increase in survival comes at a cost of reduced reproduction (Fig. 1a, solid line; convex trade-off curve)") doesn't match the figure. Survival comes at a cost of reduced reproduction in both the convex and the concave curves.

Reply: We clarified this statement (L71-74): The evolution of cell groups is, however, more likely when an increase in survival comes at a relative high cost of reproduction, where survival evolves first followed by an increase in reproduction (Fig. 1a, solid line; convex trade-off curve).

116-118: rotifer growth rate is an indirect way of estimating algal survival, but I suppose it works.

Reply: Ok. This is a good point by the reviewer, we thought so as well.

133: the enlargement constraint has not previously been described. Self-shading and reduced nutrient uptake per unit volume due to decreased surface area to volume ratio are other possible costs (or maybe this is what is meant by enlargement constraint?).

Reply: We reworded the sentence to clarify our statement. It reads now (L142-144) Potential costs for Chlamydomonas growing in cell groups include the potential for self-shading, deflagellation, loss of motility and/or sedimentation.

135-136: Figure 1D appears to show that trade-off curves differ between motile and non-motile isolates. I don't see that it shows the frequencies of swimming cells, so I don't know how it supports the difference in frequency this sentence attributes. The chi-square test does support this, but it would be nice to know what the frequencies were.

Reply: We apologize, this was a mistake. The correct reference is to Fig. 1b.

149-153: I find this section confusing. It's not clear to me how lower correlation scores imply less variable transcription profiles (I would have thought the opposite). It's also not clear what comparisons are being made here. The claim is that correlation scores are lower in cell groups than in single cells, but in both examples only one correlation score is given (so how is this a comparison?). The scores reported seem to be correlations between single cells and cell groups (lower left of the figure); the relevant scores for the claim would seem to be a comparison between single-cell correlations (upper left) and group correlations (lower right).

Reply: The reviewer is absolutely correct. We have now revised the text accordingly to (L162-165): Overall, cell groups displayed a more characteristic, i.e., less variable, transcription profile independent of the environmental conditions as we found higher correlation scores between treatments (average $\rho=0.97$) growing in cell groups than single cells (average $\rho=0.93$, Fig. 2a).

174-176: I would be interested to know how many of these variants evolved during the experiment, which could be estimated either by comparing isolates of the same starting strain across treatments (any variant found in both was probably present in the ancestor) or by comparison to pre-existing WGS data. I don't seem to have access to Table S3, so I don't know what strains were used in the experiment, but many of the Chlamy Center strains have been sequenced by Jang et al. (<https://doi.org/10.1371/journal.pone.0041307>) and by Craig et al. (<https://doi.org/10.1111/mec.15193>).

Reply: Table S3 is part of the longer SI document. As we focus on the variants that are either shared by only the cell groups or the single cells, all variants we discuss are de-novo. We can however not state whether the variant in the subset of the cell groups or the single cells evolved de-novo. We clarified this in the text by adding (L203-205): Because we compare all isolates to the same reference genome, we cannot distinguish whether a variant evolved de-novo in the cell groups or in the single cells.

183-197: it might be useful to compare genetic variants, gene expression differences, and functional assignments of both to a previous genetic analysis of experimentally evolved multi-celled groups of *C. reinhardtii* (<http://dx.doi.org/10.1098/rsos.180912>).

*Reply: This is an excellent point by the reviewer and we added this comparison to the results section; see L180-194: We further compared the differentially expressed genes to a recent study with *C. reinhardtii* and rotifers³² looking at the gene expression over time (3h, 6h, 9h, 12h and 48 hours) of an evolved isolate that is characterized by unicellular propagules from which multicellular clusters grow. Specifically, we compared differentially expressed genes from Ref 32 detected for each life cycle phase (growth lag-phase -> unicellular propagule -> multicellular cluster) to our dataset of differential*

expressed genes identified in cell groups when compared with single cells. We found fewer genes in common between the lag-phase of the life cycle (3h) and identified in our cell group selection (352 genes), while several more genes were in common at the life cycle phase of development of multicellular clusters (686 genes at 12h and 731 genes at 48h). In total there were 174 genes that were differentially expressed both in cell groups in our data set and in all multicellular induction phases in Ref 32; 64 of those genes were downregulated and 110 upregulated in cell groups and throughout the multicellular phases while 110 were upregulated in cell groups and throughout the multicellular phases (see gene list in Table S4).

Please also find in the discussion (see L277-279): Indeed, we found common expression changes between cell groups and those previously described when transition to multicellularity life cycle was induced³¹, hinting to potential commonalities in impacted metabolic pathways.

229: volvocine is not a formal taxon and does not need to be capitalized.

Reply: Thanks. We changed this.

231: "aggregates" is often contrasted with clonally developing cell groups (such as *Gonium* and *Pandorina*), so it's probably not a good choice to describe them.

Reply: We changed it to cell groups.

233: the abbreviation ECM is only used a couple of times and is probably unnecessary.

Reply: Ok, we changed this.

242: reference 35 does not report evolution of cell groups in response to predation.

Reply: We corrected this.

247-248: I would say "in some other studies", since this is not true of refs 8, 9, 12, and 35.

Reply: The reviewer is right, and we have changed it accordingly.

258: reference 12 does not report gene expression studies (genetics of that experiment are reported in <http://dx.doi.org/10.1098/rsos.180912>).

Reply: Thanks for pointing it out. We corrected the citation.

262: in the discussion of Duf3707, it may be worth mentioning that expansions in the pherophorin gene family are one of only a handful of large-scale differences between the *Chlamydomonas* and *Volvox* genomes, and these genes have been suggested to have played important roles in the evolution of multicellularity and the expansion of extracellular matrix in the volvocine algae.

Reply: We added this to the discussion. Thanks for point it out it is an important add-on to our arguments (L308-310). The pherophorine gene family has been suggested to play a major role in the evolution of multicellularity and the expansion of the extracellular matrix in the volvocine alge (refs).

299-300: I'm not sure what message the authors intend with "the existing genomic and phenotypic repertoire of the cells has been used to respond to selection." This seems self-evident and trivial, which is probably not the intent for the final sentence of the discussion. Don't all responses to selection use the existing genomic and phenotypic repertoire of the organisms under selection? I suspect that the authors can draw a stronger conclusion from the striking repeatability of their experimental populations.

Reply: We changed this to (L346-348): This fairly high degree of repeatability and the small number of generations suggest some degree of determinism for the phenotypic and genomic response in C. reinhardtii to predation pressure.

307-308: the provenance of the rotifers is vague. Where did the stock cultures originate? How were they confirmed to be B. calyciflorus?

Reply: We added information to the stock culture of B. calyciflorus. This stock culture has been used in many previous experiments: The rotifer stock culture is derived from an isolate from Milwaukee harbor, Wisconsin⁵⁰.

309-310: on what were the rotifers fed before starvation? Is 24H enough, or is it possible that some of the detected contamination was carried over food from before the experiment?

Reply: The rotifer stock culture is maintained on Monoraphidium, so any contamination after 24hrs is clearly visible. We added the information to the methods section: and maintained on Monoraphidium minutum

The length of the experiment is reported as six months and as ~500 generations, but it's not clear to me how the generation time was estimated.

Reply: We apologize for not providing more information on our calculation of the number of generations. We added to the methods: The number of generations was estimated using a doubling time of 0.4 days (~9 hours) and the number of days of the selection experiment. This calculation is based on optimal growth conditions as exact conditions in the selection experiments are not known (changes in nutrient concentrations over time due to consumption by the algal cells and bacteria as well as recycling of nutrients through rotifer and bacterial excretion).

In the supplemental spreadsheets, I'm not sure what homozygous and heterozygous mean in the context of a haploid organism.

Reply: Thanks for finding this error. We have deleted the row with the zygosity. During the reevaluation insertions and deletions were declared as heterozygous. This is probably an output error within the software. Thanks for the hint.

Matt Herron

Reviewer #3 (Remarks to the Author):

In the manuscript 'The evolution of convex trade-offs enables the transition towards multicellularity' Bernandes et al report the results from an experimental evolution study, where the green algae *Chlamydomonas reinhardtii* and the rotifer *Brachionus calyciflorus* were co-evolved. The authors report the evolution of a convex trade-off between reproduction and survival for cell groups, which has previously been predicted to facilitate the differentiation into soma and germ-like cells. Further, gene expression analysis and whole genome sequencing is employed for a subset of evolved isolates to compare gene expression patterns and genomic changes between single cells and cell groups.

The topic is timely and I very much like the approach of the study. Overall, in particular the methods part lacks significant detail that hinders replicating this study.

We thank the reviewer for the positive evaluation. We hope we could clarify all points raised by the reviewer.

I have several comments/ questions:

1) 124-26/173-76: The transition to multicellularity does not necessarily follow the route described here – the differentiation in soma and germ-like cells does only apply for a subset of the cases of clonal multicellularity. Whereas for mammals this is true, it is not so for sponges, for example.

Reply: We agree and we changed the wording to: The evolutionary transition towards multicellular life often involves growth in groups of undifferentiated cells followed by differentiation into soma and germ-like cells.

2) 136: I am not convinced that the convex trade-off evolved in cell groups as claimed here as the authors do not present the data from the ancestral lines. Also, the convex trade-off is also in the evolved single cells. It might be a general feature of the model system in that particular environment, i.e. with predators. From the Figure legend it is not clear whether the data presented is only from the predation treatment or also includes the no predation. It would have been interesting to compare the trade off between treatments.

Reply: The reviewer is correct that we did not provide sufficient information for this interpretation. We now added additional panels and analyses to Fig. 1, showing the trade-off curves for cell-groups and single cells for the two treatments (predation, no-predation) as additional panels (panel d and e in the new figure). Figure c in the old and new figures shows all data from the two treatments combined. Interestingly and in support of our interpretation, we found a significant trade-off for cell groups but not for single cells in the predation treatment and the opposite in the no-predation treatment. For the latter, there were too few data points for the estimate of the shape parameter b for the cell groups. We added the additional information to the result section and updated the methods sections. (L137-142):

When comparing the trade-off between cell groups and single cells within the two treatments, we found a significant trade-off only for cell groups for isolates from the predation lines (cell groups: $b = -0.52 \pm 0.28$; $df = 3$, $F = 21.33$, $p = 3 \times 10^{-5}$; single cells: $df = 3$, $F = 0.0128$, $p < 0.9105$; Fig. 1d) and only for single cells in the no-predation lines (cell groups: too few points to fit model; single cells: $b = -0.47 \pm 0.12$; $df = 3$, $F = 108.07$, $p < 2.2 \times 10^{-16}$; Fig. 1e).

3) It is interesting that the authors differentiate between soma and germ line cells dependent on whether cells are part of the cell group or not. The question is whether cell groups can be assigned to be soma. If this was the case, somatic cell groups would be unable to give rise to new cell-groups – per definition they would be an evolutionary dead end. The authors do not show any data to support that cell groups actually can be regarded as soma. A study by Herron et al (2019), which also used the very same prey and predator species in an experimental evolution study, report a variety of cell groups with multicellular life cycles, which themselves give rise to new cell groups. This contradicts the assignment of cell groups as soma.

Reply: We did not want to say that we observed the separation of soma and germ line cell. We actually state the opposite in the discussion: “We found a smaller percentage of isolates that grew as a mix of groups and single cells, but our test environments did not allow for ecological scaffolding and separating fitness benefits of the soma (cell group) and germ cell-like cells (single cells)³³⁻³⁵. Whether these mixed isolates would provide a fitness benefit in environments where selection for germ and soma are temporally and/or spatially separated or whether the fitness of the group is larger than the average fitness of the germ and soma cells²⁰, needs further investigations.” We changed this to (L251-257): We found a smaller percentage of isolates that grew as a mix of groups and single cells, but it is unclear whether these represent different life stages of a cell group or parts of cell groups as observed in other studies^{8,32}. As we focus in our study on the curvature of the trade-off curve, our experimental environments were not designed for testing for ecological scaffolding and separating fitness benefits of the soma (cell group) and germ cell-like cells (single cells)³⁵⁻³⁷.

4) *C. reinhardtii* is able to form palmelloids – how do the authors differentiate between them and the evolved cell groups? Herron et al (2019) observed palmelloids among their evolved groups that originated from cells staying together (clonal origin) versus from coming together (aggregative). They also evolved a heritable clonal group morphology, so that Bernandes et al cannot claim “response to predation has been observed in other studies as a phenotypically induced responses^{6,18} rather than a heritable phenotype. Differences the observation in our and other study are likely the result of the underlying process of how the cell groups form. While cell groups in the present study are the result of growth in groups where the daughter cells fail to separate after cell division (Fig. S2), in other studies cells actively aggregated for the formation of the group of cells¹⁸.”

*Reply: We apologize for not being clearer in this part of the discussion. We did not want to say that there are no other studies showing the evolution of cell groups in *Chlamydomonas* in response to predation. We rather wanted to point out that there is also the possibility for plastic responses, which are maybe more likely to be the result of aggregation rather than not separating after division. We changed the sentence to (L285-287): Cell group formation in response to predation has been observed in some other studies as a phenotypically induced responses^{19,21} rather than a heritable phenotype.*

5) Measurement of algal growth rates as OD: I worry about reliably measuring growth rates as ODs as done here as some of the isolates are a mixture of cell groups, single cells, and predators. Also, how can one measure growth of cell groups as OD? I think that estimates of cell and group densities would need to happen as done here for the predators via counting or via image analysis as done by Herron et al (2019). I particularly worry about measuring

growth of groups but also that of cells. For cells, the OD would only provide a good estimate of growth that is comparable among the different isolates if cell size among the different isolates is the same.

Reply: The reviewer has a good point here. At the time when we ran this experiment, we were not able to use microscopic counts for all the samples and we used OD measures instead. We measured at nine sites within one well and used the average of the nine sites for the growth rate estimates. We add this information to the Methods section (L382-383): Algal clones were grown with a fixed rotifer population for three days in liquid medium (160 μ M of nitrogen) and their OD (at a wavelength of 600nm, 9 sites per well on a Tecan Infinite 200 Pro microplate reader) measured throughout.

We also provide here a figure comparing growth rates estimates from OD measures and microscopic counts for a number of different isolates of Chlamydomonas from an ongoing project where we study diversification of Chlamydomonas reinhardtii in the presence of salt. While the absolute values of growth rate estimates from OD measures are lower than the estimates of counts, there is a good correlation between the two estimates (0.92). The panels on the right present microscopic images two isolates.

6) *C. reinhardtii* faces the constraint between cell division and motility. How does this relate to the different groups with motile/non-motile cells as observed here? In particular, how can the authors exclude that they were observing different parts of the cell/ group life cycle as for example in Ratcliff et al 2013 and described by Herron et al 2018: "...milestones in the life

cycle: at 3 h, clusters appear dormant; at 6 h, unicellular propagules are actively swimming; at 9 and 12 h, many have lost motility and begun to develop into multicellular clusters; and at 48 h, clusters have reached a large size (approx. 50 cells or more).“

Reply: The reviewer raises an important point here. We cannot exclude that we observe different parts of a cell group or life cycle here. We added to the discussion (L252-254): but it is unclear whether these mixes are different life stages of a cell group or parts of cell groups as observed in other studies^{8,33}.

7) 12 colonies were randomly picked at the end of the experiment from each strain/predator combination:

- How are these representative of the composition in each population?

Reply: We are not aware of standards for determining the number of re-isolates for selection experiments to capture a representative diversity of the population. The lowest frequency we can detect with 12 colonies is ~0.1 and low frequency genotypes will not be represented in our subsample analyzed here.

How closely related are they?

Reply: The 12 isolates are all derived from the same initial colony that was used to start the selection lines.

Here I worry about treating each isolate as an independent replicate, e.g. in Figure 1c, d. How do you argue against pseudo-replication? To me your experimental set up results in 10 replicates per predation/no-predation treatment. Without knowledge of the underlying relatedness within one flask, one needs to account for the particular strain/predator combination in all analyses.

Reply: The reviewer is right that it is important to take pseudo-replication into account for example using mixed effect models and treating the ancestral line as random effect. We changed the statistical analyses for the growth rates of algae and rotifers to mixed effect models. (L119-123) Growth in groups increased survival (i.e., rotifer growth rate was significantly reduced for cell groups; linear mixed effect model (LME) with strain as random effect: $F_{1,202} = 6.40$, $p = 0.012$; Fig. 1c, Fig. S2) but reduced average algal growth rate compared to single cells (LME: $F_{1,200} = 15.82$, $p = 9.74 \cdot 10^{-5}$; Fig. 1c).

- Can cells and cell-groups be distinguished due to colony morphology? If so how was the randomness of the colony choice regarding predator treatment ensured?

Reply: We did not observe difference in colony morphology on agar plates that is related single cells and cell-groups.

8) Sequencing: It is not clear which pairs of clones were chosen for sequencing. It is stated: 1333-335: “To identify potential candidate genes involved in cell groups, we sequenced whole genomes of nine pairs of clones from predation and no-predation selection experiment.” I would have expected 10 pairs of clones to be sequenced if there was one representative clone chosen per strain x predation combination. Why are there only nine?

Reply: We were not able to isolate sufficient DNA in high enough quality for sequencing for one strain. We added this information to the main text: isolates from cc2531 were not included due to insufficient DNA quality for sequencing

Also, I would have liked information on the colony morphology send for sequencing – the reader can only assume that cell groups were chosen from the predation treatment and single cells from the predation-free treatment. Nevertheless, when looking Figure 1b, the majority of isolates comprise a mixture of cell groups and single cells. There is no information on the amount of time the colonies were grown before sending to sequencing. Were the cultures checked for morphological heterogeneity, i.e. were there mixed cultures, before DNA extraction?

Reply: We added the information on the morphology of the clones to table S5 and added the information on growth conditions to the method section. We did not check morphology before DNA extractions.

9) Chemostat experiment and Transcriptome analysis: One pair of cell group and single cell isolate was chosen (I366/367). I would have liked to know why this particular pair was chosen. Did this stem from the same strain/predation combination, i.e. the same flask? What was the cell-group morphology?

Reply: We added this information. We say now (L151-154): We further tested the heritability of the evolved phenotypes by comparing the gene expression patterns for one pair single cells from the no predation lines (majority motile single cells) and evolved cell group from the predation line (groups with no single cells) using RNA-seq of the wildtype strain cc1009 (Table S3).

For the growth in the chemostats, to me it is not clear why an additional treatment, i.e. low nitrogen, was included.

Reply: The low nitrogen environment was used to simulate an environment with high competition for the limited resource. The populations in the selection experiment are expected to experienced nitrogen starvation at the end of the growth cycle and more so in predation free selection lines. We thus included this test environment in the gene expression analyses. We added this information to the main text. We say now (L156-159): with predation by rotifers (+R), with reduced nitrogen concentration to resemble low resource conditions (-N; we halved the concentration of N to 40µM) or without any of the two stressors that the populations experienced in the selection experiment (control C).

Also, the number of replicates do not add up – I am correct that 18 chemostats (and not 9) were used?

Reply: That is correct, 9 for single cells and 9 for cell groups. We replaced nine by 18.

Overall, the manuscript needs careful checking in terms of language and grammar, e.g. 151: “preyed upon” instead “fed on”; 152: “of” instead “for”; 1129: “having” instead “have”, 1367: “chemostats” instead “chemostat”....

Reply: We apologize for the many mistakes.

Other comments:

l. 32: which trait? – cell groups?

Reply: Yes, cell groups. Changed to: cell groups were

Figure S1, 110: remove: “(see STAR methods)

Reply: Done.

REVIEWERS' COMMENTS

Reviewer #1 (Remarks to the Author):

Thanks for letting me see it again. The authors have addressed my comments and I recommend it be published in Nature communications.

Small stylistic suggestion for the first three lines of the abstract at the discretion of the authors. Change to something like:

“The evolutionary transition towards multicellular life often involves growth in groups of undifferentiated cells followed by differentiation into soma and germ-like cells. Theory predicts that germ soma differentiation is facilitated by a convex trade-off between survival and reproduction. However, this has never been tested as these transitions remain poorly understood at the ecological and genetic level. Here we studied...”

Rick Michod

Reviewer #2 (Remarks to the Author):

The authors have addressed all of my comments to my satisfaction.

Reviewer #3 (Remarks to the Author):

The authors fully addressed my questions and concerns.